# Pre-Incisional and Multiple Intradermal Injection of N-Acetylcysteine Slightly Improves Incisional Wound Healing in an Animal Model

**DOI:** 10.3390/ijms25105200

**Published:** 2024-05-10

**Authors:** Wiktor Pascal, Antoni Smoliński, Mateusz Gotowiec, Marta Wojtkiewicz, Albert Stachura, Kacper Pełka, Michał Kopka, Kyle P. Quinn, Alan E. Woessner, Dariusz Grzelecki, Paweł Włodarski

**Affiliations:** 1Department of Methodology, Medical University of Warsaw, 02-091 Warsaw, Poland; s082979@student.wum.edu.pl (A.S.); mateusz.gotowiec@wum.edu.pl (M.G.); m.z.wojtkiewicz@gmail.com (M.W.); albert.stachura@wum.edu.pl (A.S.); kacper.pelka@wum.edu.pl (K.P.); michal.kopka@wum.edu.pl (M.K.); pawel.wlodarski@wum.edu.pl (P.W.); 2Doctoral School, Medical University of Warsaw, 02-091 Warsaw, Poland; 3Department of Biomedical Engineering, University of Arkansas, Fayetteville, AR 72701, USA; kpquinn@uark.edu (K.P.Q.); aewoessn@email.uark.edu (A.E.W.); 4Department of Orthopedics and Rheumoorthopedics, Centre of Postgraduate Medical Education, Professor Adam Gruca Orthopedic and Trauma Teaching Hospital, 05-400 Otwock, Poland; dariuszgrzelecki@gmail.com

**Keywords:** wound healing, skin, N-acetylcysteine, pretreatment, rat, local anesthesia additive, incision, surgical

## Abstract

The objective of this study was to investigate if delivering multiple doses of N-acetylcysteine (NAC) post-surgery in addition to pre-incisional administration significantly impacts the wound healing process in a rat model. Full-thickness skin incisions were carried out on the dorsum of 24 Sprague-Dawley rats in six locations. Fifteen minutes prior to the incision, half of the sites were treated with a control solution, with the wounds on the contralateral side treated with solutions containing 0.015%, 0.03% and 0.045% of NAC. In the case of the NAC treated group, further injections were given every 8 h for three days. On days 3, 7, 14 and 60 post-op, rats were sacrificed to gather material for the histological analysis, which included histomorphometry, collagen fiber organization analysis, immunohistochemistry and Abramov scale scoring. It was determined that scars treated with 0.015% NAC had significantly lower reepithelization than the control at day 60 post-op (*p* = 0.0018). Scars treated with 0.045% NAC had a significantly lower collagen fiber variance compared to 0.015% NAC at day 14 post-op (*p* = 0.02 and *p* = 0.04) and a lower mean scar width than the control at day 60 post-op (*p* = 0.0354 and *p* = 0.0224). No significant differences in the recruitment of immune cells and histological parameters were found. The results point to a limited efficacy of multiple NAC injections post-surgery in wound healing.

## 1. Introduction

Wound healing is a vital process for maintaining the homeostasis of an organism. It may be described as a sequence of complex molecular and cellular events that lead to the closure of a defect with scar tissue. Throughout years of research, this phenomenon has been divided into several phases, in which different molecular processes occur. First, in the inflammatory phase, immune cells are recruited to clear the wound bed from pathogens, dead cells and injured extracellular matrix (ECM). Next, proliferation commences, with the intense activation of local cells and fibroblasts to fill the deficit of tissue in the injured area. Remodeling is the final phase, where the maturation of ECM and cells occurs (Figure 1).

Since each phase is dependent on the prior one, extended or exaggerated inflammation may hamper healing, prolong the subsequent phases, or exacerbate scarring [1]. Thus, decreasing the level of inflammation or its products (reactive forms of oxygen (ROS), advanced glycation end products (AGE), overexpression of cytokines or overrecruitment of neutrophils, etc.) may improve the next phases of wound healing.

N-acetylcysteine (NAC) is a well-known antioxidant particle. Its simple chemical structure and unique properties encourage us to explore its potential for further applications. Recent clinical trials cover ischemic stroke, urology, adjunct analgesic properties or therapy of cytokine storm [2,3,4,5]. The pleiotropic effect and universal properties of NAC have been also utilized in wound healing [6,7,8]. Our previous studies concern a unique model of the pre-incisional administration of NAC to the future wound bed of a surgical wound, as an additive to local anesthesia [9,10]. Promising effects and a lack of substantial side effects encouraged us to explore whether an extension of the therapy from pre-incision to the early phase of wound healing may turn out to be even more fruitful. Recently, we showed that a single dose of NAC applied after the creation of a wound produced slightly slimmer scars with improved histology [9] followed by long-term (over a 60 day observation period) shifts in the gene expression of crucial targets for wound healing and remodeling [10]. Notably, the HIF1alpha pathway was pronouncedly expressed at all timepoints, which is consistent with the findings of other authors, and may be the basis of NAC efficiency in improving wound healing [11,12]. To date, no consecutive intradermal therapy with NAC for wound healing has been tested. Other studies focused on using hydrogels, ointments or dressings, which were mainly used for excisional wounds. Contrastingly, most of the wounds to be healed are planned, surgical incisions. Thus, exploring the improvement of the healing process or decreasing scarring in such wounds may be useful in clinical practice.

We hypothesized that the extension of NAC-based therapy to the early phase after injury may provide a further decrease in the extent of inflammation and an improvement in wound remodeling.

In this study we aimed to verify if the prolongation of the pre-incisional therapy with further intradermal injections of NAC near the scar may further improve the healing of a surgical wound in a rat model.

## 2. Results

### 2.1. Histomorphometry

No statistically significant differences (mixed effects random model analysis, *p* > 0.05) between the control group and NAC15/NAC30/NAC45 groups were found at any time point in terms of the histomorphometrical parameters of the wound (Figure 2). The n-value differed from each category due to the occasional low quality of the samples, which made the acquirement of precise results impossible. Raw data can be retrieved in Appendix A.

The NAC30 group had a lower GRI and MRI score (*p* = 0.06; *p* = 0.06), but a higher area of connective tissue in the wound center at day 7 post-op compared to the control (*p* = 0.06). The thickness of the epidermis (EPI) was higher in the control at day 7 post-op in comparison to NAC15/30/45, however not significantly (*p* > 0.05)

### 2.2. Collagen Fiber Organization

The characteristics of collagen fibers in the scar and in areas both distal and proximal to it were measured in terms of overall/local directional variance, density and blue color intensity at the 3rd, 7th, 14th and 60th day post-op (Appendix A). Raw data can be found in Appendix A. There was a statistically significant decrease in the directional variance of collagen fibers in the NAC45 group compared to NAC15 in distal scar areas (RD&LD) and the scar (S) at day 14 post-op (Figure 3, *p* = 0.02 and *p* = 0.03, respectively). There were no statistically significant (ANOVA, *p* > 0.05) differences in any other outcomes at any time points.

### 2.3. Immunohistochemical Staining

There were no statistically significant differences (mixed-effects model with post hoc Bonferroni’s multiple comparisons test, *p* > 0.05) between the control and NAC15/NAC30/NAC45 groups for any variable at any time point post-op in terms of immunohistochemical staining. Thus, further analysis was focused on the comparison between combined NAC15/NAC30/NAC45 (gNAC) and control group (Table 1).

### 2.4. Histological Assessment of Scars

Histological assessment was performed using digitalized MT- and HE- stained scar sections. Analyzed, representative samples at every post-op interval are displayed in Figure 4. There were no statistically significant differences (Kruskal–Wallis test, *p* > 0.05) between the mean values of any of the measured variables for NAC15/NAC30/NAC45 and the control at days 3, 7 and 14 post-op. Statistically significant differences were observed in reepithelization at day 60 post-op among studied groups (Kruskal–Wallis test, *p* < 0.05). Results obtained from the variables at all the studied points are displayed in Figure 5.

When conducting post hoc analysis, it was found that the differences in the mean values for reepithelization between the NAC15 and control groups at day 60 post-op were statistically significant (Mann-Whitney test, *p* = 0.0018). The mean value of the NAC15 reepithelization evaluation score was significantly lower than that of the control (Figure 6).

### 2.5. Wound and Scar Planimetric Analysis Based on Photographic Documentation

The mean scar width showed a significant effect on NAC groups (fixed effects of the mixed-effects analysis model and chi-square, *p* = 0.0044 and *p* < 0.001, respectively, Figure 7A). The mean scar width of the NAC45 group was the lowest on the 60th day of observation post-op compared to other groups at the same time point. In the post hoc analysis of subgroups, it was shown that the difference between the mean scar width of NAC45 and the control at day 60 post-op was statistically significant (Bonferroni’s multiple comparisons test and Mann–Whitney test, *p* = 0.0354 and *p* = 0.0224, respectively, Figure 7B).

The scar length was shown to be statistically significantly influenced by NAC treatment (Fixed effects of the mixed-effects analysis model and Chi-square, *p* = 0.0034 and *p* < 0.001, respectively, Figure 7C) with the mean of the NAC15 group being the lowest at almost all time points, except for the 21st and 45th day of observation post-op. Despite that, none of the specific NAC groups displayed any significant difference when compared with the control group (Bonferroni’s multiple comparisons test, *p* > 0.05).

There was no statistically significant effect of NAC treatment on the mean values of scar area at any time point (Fixed effects of the mixed-effects analysis model, *p* = 0.11, Figure 7D). The mean lowest values of this parameter were in the control group at most time points and the highest in NAC30 (Bonferroni’s multiple comparisons test, *p* > 0.05).

## 3. Discussion

Our results concerned the histomorphometry, collagen fiber organization, immunohistochemical staining, and the histological and planimetric assessment of wound healing. In terms of histological parameters, we found no significant impact of multiple-dose NAC administration at any concentration. In our precursory study, we determined the influence of applying single doses of NAC of the same concentrations (0.015%, 0.03% and 0.045%) on the efficacy of wound healing. In the previous study, the superficial contraction index and dermal proliferation area were significantly (*p* < 0.05) higher after 0.03% NAC application compared to the control at day 7 post-op, whereas in this study, the aforementioned values were not significantly higher in any tested group. The values which were closest to being statistically significant, namely the Global Remodeling and Matrix Remodeling Indices (GRI; MRI) and epidermal thickness for the 0.03% NAC group all suggest that multiple dose application is detrimental compared to single doses. The GRI decreased by 1.61 compared to the control at day 60 post-op, whereas in our previous study it decreased by only 0.13, and the MRI decreased by 2.75 compared to 0.14 previously. Interestingly, the epidermal thickness at day 60 post-op also decreased by 1.27 µm (*p* > 0.99), whereas in the single-dose study it increased by 0.3 µm [9]. As such, it can be said that applying multiple doses of NAC to wounds is not only not superior to the single-dose method, but in the case of the 0.03% NAC dose seems to be disadvantageous in terms of improving histomorphology.

We calculated the density, intensity and variance of fiber directions around the NAC administration area and in different parts of the wound/scar area using an automated analysis of collagen fiber properties. We found that 0.045% NAC treatment resulted in a decrease in overall directional variance compared to 0.015% NAC, but not in the control at day 14 post-op. Based on these results, an assumption can be made when considering the dose–response to N-acetylcysteine in the early wound healing period (up to day 14 post-op). It seems that 0.045% NAC may be beneficial by promoting faster collagen reorganization, while 0.015% NAC causes a more chaotic development of collagen fibers. Such results correspond to data gathered by Zhou et al. [13], who have shown that NAC may not only have anti-oxidative effects, but also limits collagen over-synthesis by fibroblasts. We did not observe any significant differences in collagen properties in the late-stage scar formation period (60th day post-op) following multiple-dose NAC administration. This corresponds to our previous work [10], where we reported no long-term effects of single-dose NAC on collagen fiber properties. When comparing specific values between our studies, multiple-dose administration increased the fiber directional variance and decreased fiber density compared to single doses. This may be related to the damage caused by multiple injections, which leads to a more intense remodeling of the scar.

Using immunohistochemical staining, we explored the influence of NAC on the inflammatory process within the wound itself. Antigens CD68 and MPO were present, which are commonly found on macrophages and neutrophils, respectively. However, there were no statistically significant differences between the number of CD68-positive and MPO-positive cells in wounds treated with NAC and the control. This may suggest that multiple doses of NAC have no significant influence on the number of both macrophages and neutrophils migrating to the wound site, thus not promoting an enhancement of the wound healing process [14,15]. These results contradict our previous study [10], which showed a significant increase in CD68-positive cells in wounds treated with a single dose of 0.03% NAC at day 14 post-op. This may indicate the potential superiority of single-dose treatment over multiple doses in the wound healing process. Antigen CD31 was also detected and is found mainly on endothelial cells, but also on platelets and most subtypes of leukocytes [16]. As such, it acts as a marker for neovascularization. We observed that there was a decrease (*p* > 0.05) in the number and percentage of CD31 positive cells present in wounds treated with multiple doses of NAC in comparison to the control (with the exception of day 7 post-op), which is contrary to our previous findings [10], which showed that a single dose of 0.03% NAC significantly increased the number and percentage of positive cells with this antigen present within the wound at day 14 post-op. This further shows that a single pre-incisional dose of NAC can be considered more beneficial than multiple doses when it comes to processes like neo-angiogenesis [17].

Our investigation also involved the application of subjective histological assessments using the modified Abramov scale to compare variances in wound healing process between NAC-treated wounds and the control cohort. No significant differences surfaced across any category between multiple doses of NAC, regardless of concentration, and the control group on days 3, 7, and 14 post-op. At day 60 post-op, we observed that multiple doses of 0.015% NAC negatively impacted the reepithelization of scars in contrast to the control (*p* < 0.05). This observation suggests a potential impediment to the development or maturation of the epidermal layer, an important aspect in the reparative cascade of both the epidermis and dermis. This may lead to the potential exacerbation of the wound. This contradicts our previously published findings [9], which showed that a single dose of NAC at any concentration had no significant impact on any of the assessed histological categories at the same time points. This further supports the claim of the superiority of a single dose of NAC over multiple doses. Moreover, the proposition arises that the administration of NAC via multiple injections could have led to the creation of microinjuries, additional irritation, oedema, pain and damage to the surrounding tissue at the injection site [18], which could lead to a potential increase in the morbidity of the studied subjects. In addition, microinjuries to the underlying dermis could have contributed to notable adverse outcomes seen in the studied categories. Additionally, the concentrations reached using multiple injections may have been different than those achieved with single dosing, which could potentially explain the contradictory results observed in our previous report. These conclusions raise a question about the method of NAC application. Notably, Tsai et al. [6] have found that topical NAC application could have beneficial effects on reepithelization and may be the preferred method of administration, although this requires further investigation.

Lastly, we investigated wounds using planimetric analysis based on photographic documentation. In the case of scar area, we did not find any statistically significant differences between the control and NAC treated wounds. However, it is noteworthy that the mean scar area for wounds treated with 0.03% NAC tended to be higher across most examined time points. This observation reinforces the findings from immunohistochemical staining and subjective histological assessments, suggesting that 0.03% NAC via multiple injections may be the least efficacious concentration among those studied, and may even hinder the wound healing process. This contrasts our prior publication [9], which found that 0.03% NAC given as a single dose significantly decreases scar area on day 3 post-op. Moreover, our analysis revealed that scar length was statistically impacted by NAC treatment (*p* = 0.0044 and *p* < 0.001), however with further analysis we did not find any statistically significant differences between NAC treated groups and the control. Nevertheless, we observed that mean scar length was almost always the lowest (*p* > 0.05) in the case of 0.015% NAC, with the exception of the 21st and 45th day post-op. Thus, it may suggest a higher wound contraction compared to the other groups. Along with the observation that those scars had a decreased level of reepithelization, we may assume that scars had a tendency for superficial atrophy, which may produce less durable scars. However, there were no signs of atrophy in the dermal layer.

During the analysis of mean scar width data, we found a statistically significant decrease at day 60 post-op in 0.045% NAC (*p* = 0.0354 and *p* = 0.0224) in contrast to the control. This finding contrasts our prior report [9], in which 0.03% NAC given in a single dose significantly decreased scar width. This prompts further inquiry into whether a significant improvement in the wound healing process observed in wounds treated with multiple NAC injections could be attributed to higher concentrations as opposed to lower ones.

Intriguingly, significant differences in the studied parameters were not observed in the early days post-op in wounds treated with multiple NAC administration, nor in the control. This contrasts our previous study [9], in which there were statistically significant differences in mean scar area on day 3 post-op and mean scar width on day 4 post-op. In the previous study [9], there was a significant difference in mean scar areas between 0.03% NAC and the control, which was 20.5 mm^2^ and 24 mm^2^, respectively, at day 3 post-op. In this study, on day 3 post-op, all the values for the mean scar area oscillated at around 23 mm^2^, with a non-significant decrease in 0.03% NAC seen on day 4 post-op. When it comes to scar mean width at day 60 post-op, in our previous study [9], 0.045% of NAC had the biggest scar mean width (approx. 2.4 mm), while in this investigation 0.045% of NAC had the lowest value out of all studied concentrations (approx. 1.8 mm). This may suggest that a single dose of NAC has a significant effect during the inflammatory phase, while multiple administrations impact the advanced remodeling phase of the wound healing process.

Considering these results, further research is required to explore the relation between dose and effectiveness of multiple NAC injections on wound healing. Furthermore, using a different wound healing model could provide some explanation for the outcomes. This study used an incisional model of wound healing, which has a higher translational context, as it corresponds to primary adhesion healing, occurring commonly in uncomplicated wounds, but may not show the benefits of NAC injections in chronic, contaminated or ischemic wounds. Pranantyo et al. [19] have shown that the co-administration of NAC and other antioxidative/antibiofilm substances embedded in a hydrogel accelerated the infected wound closure in a diabetic model. Ozkaya et al. [20] have shown promising results regarding combinational treatment using both the systemic and topical administration of NAC in a diabetic rat model. In both studies, an excisional model of wounds was used, and both showed significantly better outcomes following NAC administration. Similarly, Hou et al. [21] have reported faster wound closure and better epidermal maturation with sustained NAC release using polycaprolactone/collagen scaffolds. The study also used an oval full-depth wound model, showing the potential benefits of long-term NAC administration using different administration vehicles, wound models and clinical contexts.

All the data gathered and a thorough examination of the wound healing process indicate that the 0.045% dose of NAC slightly decreases scar width and collagen fiber variance in the early and mid-phases of wound healing. However, the effects were limited.

This underscores the necessity for further research, which would lead to a better understanding of the dose–effect relationship and determine the most effective method of NAC administration.

## 4. Materials and Methods

### 4.1. Animals

The following protocol is a modified version of a previous report undertaken by our group [9,10]. The experiments were approved on 26 April 2017 by the First Local Ethics Committee in Warsaw (Protocol no. 304/2017), and the study was conducted according to ARRIVE and EU Directive 2010/63/EU guidelines. The rats were obtained from the Central Laboratory of Experimental Animals of the Medical University of Warsaw (license no. 037). As previously mentioned, inbred male Sprague-Dawley rats aged 10–12 weeks (n = 24) were acclimatized and housed in a specific pathogen-free room at the Central Laboratory of Experimental Animals, Medical University of Warsaw. In the post-operative period, the rats were housed in separate cages with environmental enrichment to avoid wound contamination and biting by cohabitants.

### 4.2. Surgical Procedures

The rats were anesthetized with an intraperitoneal injection of ketamine (100 mg/kg bw; Ketamina, Biowet, Pulawy, Poland) and xylazine (10 mg/kg bw; Xylapan, Vetoquinol Biowet, Gorzow Wielkopolski, Poland). Each of the 24 rats had six incisions carried out on the dorsal side, and the template was used to obtain identical incisions at identical distances.

The sides (left and right) were randomly assigned to the control groups (CONT) or experimental groups (NAC). One side received local anesthetic (0.5% lidocaine (Lignocainum Hydrochloricum WZF, Polpharma, Warsaw, Poland) and a vasoconstrictor (1:100,000 epinephrine (Adrenalina WZF, Polpharma, Warsaw, Poland), at a total volume of 0.6 mL for each planned incision for the control group. The remaining side was treated with 3 different concentrations of NAC (Acetylcysteine, Sandoz, Warsaw, Poland): 0.015%—NAC15 group, 0.03%—NAC30 group, and 0.045%—NAC45 group, dissolved in the anesthetic solution as in the control group. Injections were randomly assigned to rostral, central or caudal incision. They were performed by a blinded researcher. NAC dosage was calculated as in previous studies [9,10]. For multiple administrations, doses were maintained accordingly to the random assignment of pre-incisional injection.

Dorsal skin was disinfected and each site received six intradermal injections (Figure 8) with a 30 G needle (Sterican, B.Braun, Melsungen, Germany) 15 min prior to incising the skin. Whole thickness skin incisions (1.5 cm each) were performed with fresh blade no. 11 (ZARYS International Group, Zabrze, Poland). Wounds were closed with two 4-0 Prolene (Ethicon, Johnson & Johnson, New Brunswick, NJ, USA) horizontal mattress sutures. The surgeon was blinded to the pattern of injected solutions. After surgery, the animals received an intramuscular injection of penicillin (100,000 IU/kg bw; Penicillin, Polpharma, Warsaw, Poland) and intraperitoneal injections of buprenorphine chloride (0.3 μg/kg bw; Bupaq Multidose, Richter Pharma, Wels, Germany) during the first postoperative day. Further injections of NAC solutions were performed every 8 h for 3 days following the surgery. The injection pattern was consistent with the previously reported, pre-incisional administration scheme [9,10]. Sutures were removed on the 7th post-op day.

### 4.3. Evaluation of Macroscopic Wound Healing in Time—Photographic Documentation and Scar Area Quantification

Similarly to previous studies, photographic documentation of wounds was taken on days 1, 2, 3, 4, 7, 14, 21, 28, 35, 45, 60 after the surgery. The procedure was standardized, and performed with a high resolution camera with identical light exposure and capturing parameters. Each wound was photographed with a microscale. Photos were uploaded to ImageJ 1.48v. software (National Institutes of Health, Bethesda, MD, USA) by a blinded researcher who measured 3 basic dimensions—length, width and area of each scar (in triplicate) (Appendix A).

### 4.4. Scar Tissue Collection, Staining and Analysis

On days 3, 7, 14, or 60 after the operation, six randomly chosen rats were sacrificed. Scars were excised and preserved for histologic analyses. The central part of the scar was preserved in 10% formalin solution for histologic, Trichrome and immunohistochemistry staining. Specimens were paraffin-embedded, sectioned (3–5 μm) and HE-stained in a standard protocol in the Department of Pathology of Medical University of Warsaw, as previously mentioned. Masson’s Trichrome (MT) (Sigma-Aldrich, Saint Louis, MO, USA) staining was performed according to the manufacturer’s guidelines.

After deparaffinization, immunohistochemistry staining was performed with an EnVision™ FLEX Mini Kit, High pH (DAKO, Agilent, Santa Clara, CA, USA). Anti-CD68 (ED-1, DAKO, Agilent, Santa Clara, CA, USA) clone KP1, IgG1κ, (IR609, DAKO, Agilent, Santa Clara, CA, USA), anti-MPO (myeloperoxidase) IgG1κ (IR511, DAKO, Agilent, Santa Clara, CA, USA), and anti-CD31 clone JC70a, IgG1κ (IR610, DAKO, Agilent, Santa Clara, CA, USA) were used for staining macrophages, neutrophils and blood vessels according to previously used method [22]. Immunohistochemical staining was performed with Autostainer Link48 (DAKO, Agilent, Santa Clara, CA, USA).

All stained sections were scanned at a 40× magnification in NanoZoomer XR C9600-12 (Hamamatsu, Iwata City, Japan).

WSI (Whole Slide Images) underwent automated analysis in QuPath v.0.1.2 [23] with a cell detection tool, optimized for the specimens stained in the study. Results are presented as the percentage of positively stained cells and the absolute numbers of stained cells per 1 mm^2^ of tissue.

### 4.5. Scar Histology Analysis

Digital scans of HE- and MT-stained scar sections underwent manual histologic assessment. A semi-quantitative, blinded evaluation was used, using the modifed Abramov scale [24] (Appendix A).

### 4.6. Histomorphometry

HE- and MT-stained sections were assessed with the wound healing histomorphometric parameters proposed by Lemo et al. [25] and by Gouma et al. [26] (Appendix A). Measurements were performed on WSI by a blinded researcher in QuPath.

### 4.7. Collagen Fiber Arrangement Analysis

Digital images of MT-stained sections were analyzed to quantify collagen fiber organization, as previously described by Quinn et al. [27,28] (Appendix A). Similarly, 5 regions were determined to analyze the configuration of collagen fibers—scar/wound, and the proximal and distal regions of both sides of the scar/wound (Appendix A).

### 4.8. Statistical Analysis

Statistical analyses were explicitly adjusted to each studied category based on the amount of data gathered. Mixed-effects random model analysis, a two tailed ANOVA with post hoc Bonferroni’s multiple comparisons test, a mixed-effects analysis model, a chi-square with post hoc Bonferroni’s multiple comparisons test, a Mann–Whitney test, and a Kruskal–Wallis test with post hoc Mann–Whitney test were performed in GraphPad Prism 10.2.1 (GraphPad Software, Boston, MA, USA) with the statistical significance threshold being set at *p* ≤ 0.05.

## 5. Conclusions

NAC doses administered via multiple injections in the early phase of surgical wound healing showed no significant improvement in the investigated parameters of immunohistochemistry and the histological assessment of overall wound healing processes.Multiple doses of 0.045% NAC significantly decreased mean scar width at the late stage of the wound healing process (day 60) and improved collagen fiber organization in the early and remodeling phases.It is recommended for future studies to combine the pre-incisional NAC intradermal injections with further doses administered topically to minimize injury to the dermal layer.It is proposed that NAC therapy may be administered in different phases of wound healing as an adjunctive therapy, as it significantly lowers the mean width of the surgical scar.

## Figures and Tables

**Figure 1 ijms-25-05200-f001:**
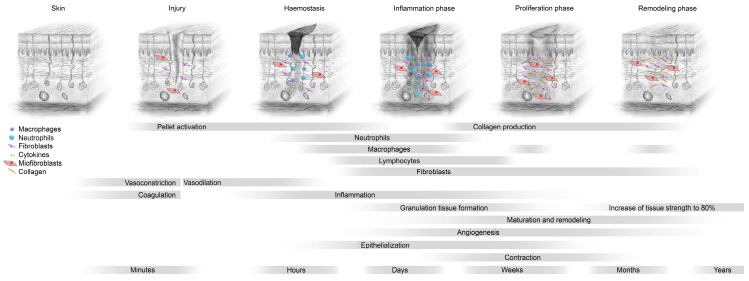
Graphical representation of subsequent phases of wound healing.

**Figure 2 ijms-25-05200-f002:**
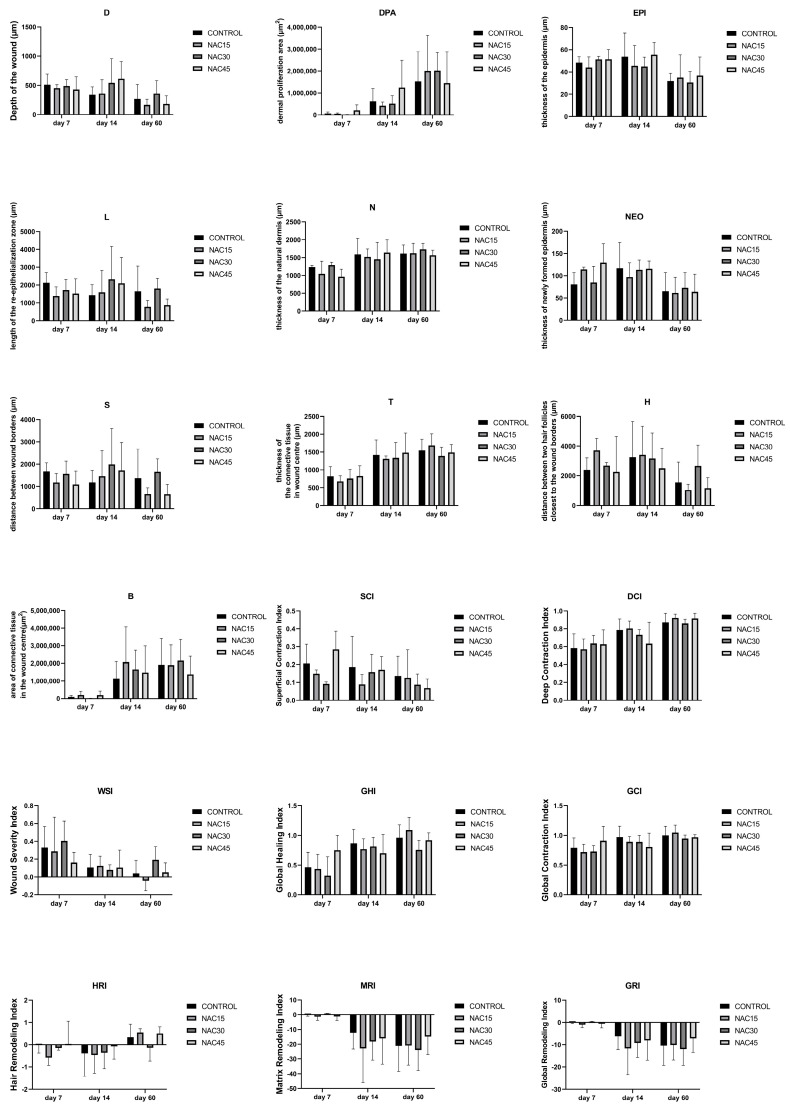
Graphs depicting histological parameters of wounds, (n = 72–81). Data expressed as mean ± SD. D—depth of the wound, DPA—dermal proliferation area, EPI—thickness of the epidermis, L—length of the reepithelialization zone, N—thickness of the natural dermis, NEO—thickness of newly formed epidermis, S—distance between the borders of the wound, T—thickness of the connective tissue in the center of the wound, H—distance between two hair follicles closest to the wound borders, B—dense scar tissue compartment, SCI—superficial contraction index, DCI—deep contraction index, WSI—wound severity index, GHI—global healing index, GCI—global contraction index, HRI—hair remodeling index, MRI—matrix remodeling index, GRI—global remodeling index. None of the differences between values were statistically significant.

**Figure 3 ijms-25-05200-f003:**
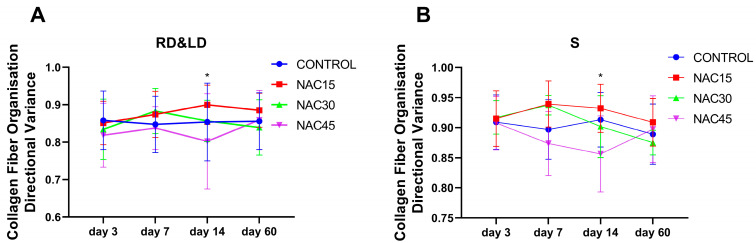
Graphs showing directional variance of collagen fibers on TM-stained sections collected from distal scar areas (n = 286) (**A**) and from the scars (n = 144) (**B**) at 3rd, 7th, 14th and 60th days post-op. Values expressed as mean ± SD. *—t-test, *p* < 0.05 for NAC15 vs. NAC45. None of the rest of the differences between values were statistically significant.

**Figure 4 ijms-25-05200-f004:**
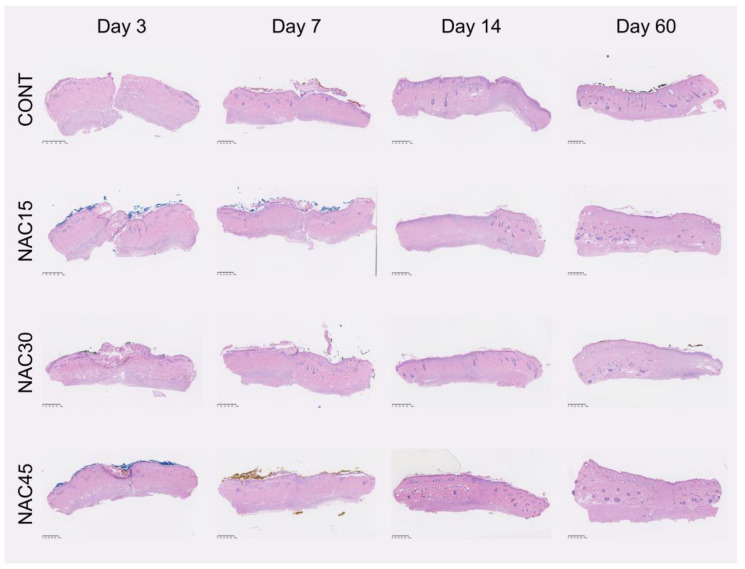
Representative samples of HE sections from each group (NAC15, NAC30, NAC45, CONT) at every sampling interval post-op. At the bottom left of each sample, a 1 mm scale bar is provided. Different colors on the edge of epidermis were derived from histopathological ink used for sample identification.

**Figure 5 ijms-25-05200-f005:**
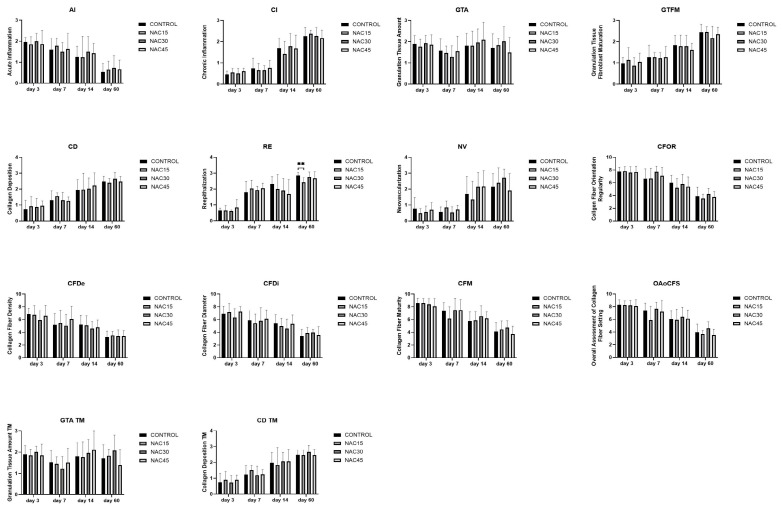
Graphs presenting results of assessed variables at all time points post-op obtained using histological analysis with Abramov scale, (n = 141). Mean values are shown. Error bars are SD. AI—acute inflammation, CI—chronic inflammation, GTA—granulation tissue amount, GTFM—granulation tissue fibroblast maturation, CD—collagen deposition, RE—reepithelization, NV—neovascularization, CFOR—collagen fiber orientation regularity, CFDe—collagen fiber density, CFDi—collagen fiber diameter, CFM—collagen fiber maturity, OAoCFS—overall assessment of collagen fiber setting, GTA TM—granulation tissue amount based on analysis of Masson’s Trichrome-stained images, CD TM—collagen deposition based on analysis of Masson’s Trichrome-stained images. Results of assessment expressed as mean ± SD; **—*p* < 0.01, None of the rest of the differences between values were statistically significant.

**Figure 6 ijms-25-05200-f006:**
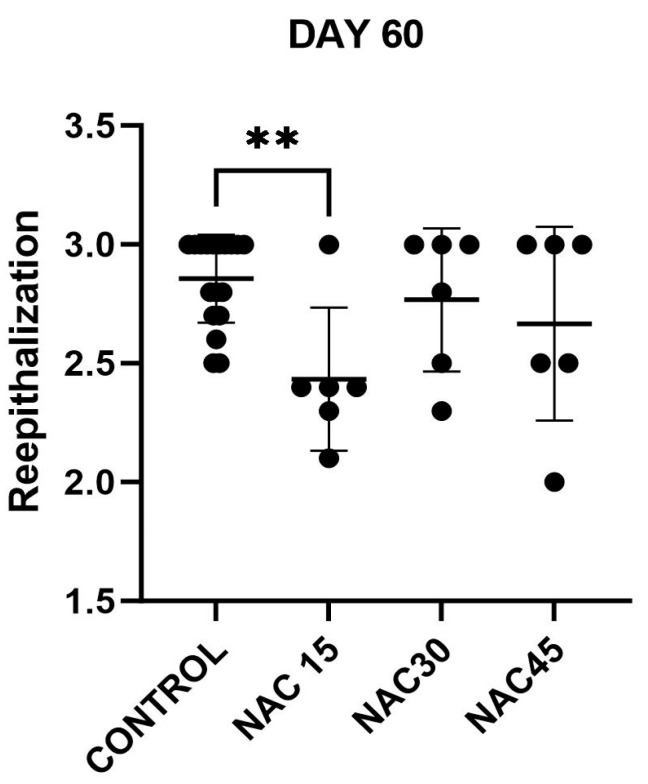
Figure displaying reepithelization evaluation scores at day 60 post-op in control group and NAC15, NAC30, NAC45 groups, respectively, with differences between groups measured using Mann–Whitney test, (n = 36), **—*p* < 0.01. None of the rest of the differences between values were statistically significant.

**Figure 7 ijms-25-05200-f007:**
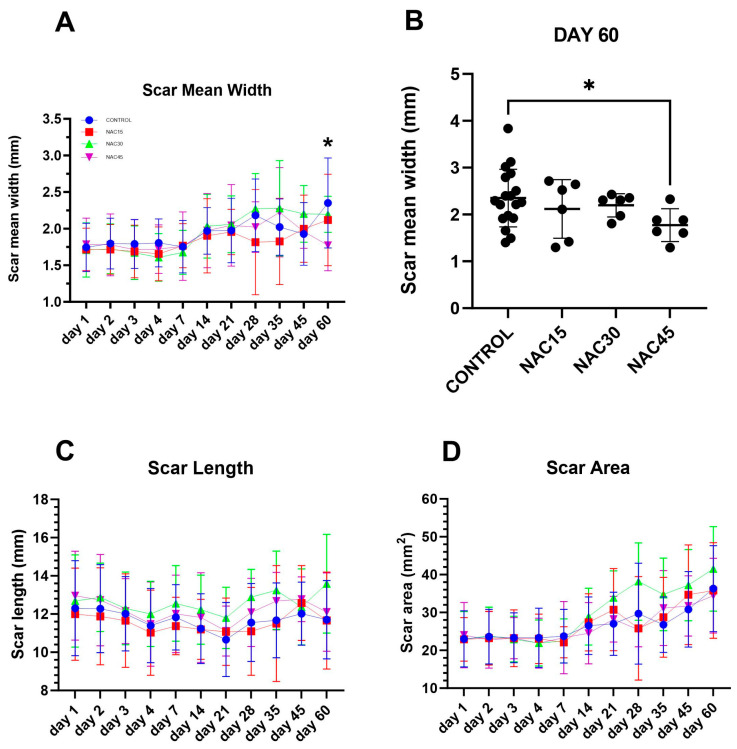
Graphs representing data of the planimetric measurements of photographed scars, (n = 144). (**A**) scar mean width at days 1–60 post-op, (**B**) scar mean width at day 60 post-op in control group and NAC15, NAC30, NAC45 groups, respectively, with differences between groups measured using Mann–Whitney test (n = 36), (**C**) scar lengths at days 1–60 post-op, (**D**) scar areas at days 1–60 post-op. Mean results ± SD. *—*p* < 0.05. None of the rest of the differences between values were statistically significant.

**Figure 8 ijms-25-05200-f008:**
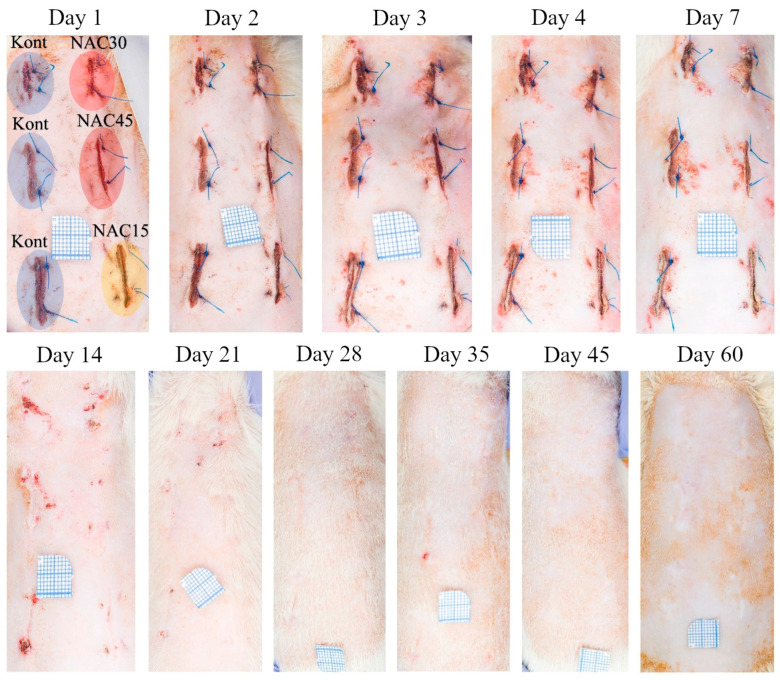
Photographs depicting the wound healing process captured at all 11 time points of the 60 day observation period post-op in a representative rat.

**Table 1 ijms-25-05200-t001:** Results obtained from immunohistochemical staining with anti-CD31 (n = 138), anti-CD68 (n = 144) and anti-MPO (n = 144) antibodies at a studied time point. Results of groups NAC15, NAC30 and NAC45 collectively analyzed as gNAC. Data expressed as mean ± SD; t-test value for gNAC vs. Control; *n*-number. None of the differences between values were statistically significant.

	Day 3			Day 7			Day 14			Day 60		
	Control	gNAC	*p*	Control	gNAC	*p*	Control	gNAC	*p*	Control	gNAC	*p*
	Mean ± SD	Mean ± SD		Mean ± SD	Mean ± SD		Mean ± SD	Mean ± SD		Mean ± SD	Mean ± SD	
CD31% of positive cells	13.61 ± 4.95	14.83 ± 5.84	0.50	4.07 ± 3.60	4.06 ± 2.38	0.99	4.69 ± 3.35	3.72 ± 2.31	0.32	2.32 ± 1.10	2.08 ± 0.99	0.48
CD31 *n* of positive cells/mm^2^	284.79 ± 116.89	261.83 ± 118.359	0.56	92.14 ± 95.45	95.57 ± 69.35	0.90	104.18 ± 88.28	98.43 ± 80.88	0.84	49.58 ± 26.73	42.87 ± 26.91	0.46
CD68% of positive cells	13.22 ± 4.66	12.89 ± 4.83	0.85	4.57 ± 2.84	6.51 ± 6.31	0.24	8.54 ± 10.14	7.37 ± 12.05	0.76	1.31 ± 1.24	1.01 ± 0.58	0.36
CD68 *n* of positive cells/mm^2^	485.88 ± 155.56	487.91 ± 189.11	0.97	165.24 ± 125.13	232.01 ± 215.72	0.26	369.41 ± 459.21	310.91 ± 517.73	0.72	46.01 ± 46.19	34.39 ± 21.52	0.34
MPO% of positive cells	31.20 ± 5.59	22.96 ± 4.79	0.65	4.41 ± 5.03	4.41 ± 3.73	0.99	3.96 ± 3.07	3.85 ± 4.88	0.93	1.44 ± 0.91	2.06 ± 2.46	0.32
MPO *n* of positive cells/mm^2^	461.19 ± 194.93	456.33 ± 206.25	0.94	157.43 ± 189.01	166.50 ± 156.83	0.88	163.42 ± 139.78	170.23 ± 231.46	0.92	48.61 ± 30.86	71.20 ± 80.63	0.27

## Data Availability

Raw data are available upon request to the corresponding author (W.P.).

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
