# Peer review of "Pre-Incisional and Multiple Intradermal Injection of N-Acetylcysteine Slightly Improves Incisional Wound Healing in an Animal Model"

_ijms, 2024, doi:10.3390/ijms25105200_

Round 1
Reviewer 1 Report
Comments and Suggestions for Authors
Comments
1- Abstract should be mentioned with more detail and include significant results.
2- Improve the resolution and quality of figure 1 and figure 4.
3- Include relevant reference(s) in lines 54-55
4- Include relevant reference(s) in lines 66-67
5- In figure 2, the mean depth of the wound on day 14 increased relative to day 7 in treated NAC30 and NAC45, why?!
6- In figure 2 and figure 4 include statistical difference on all panels between groups on specific day.
Reviewer 2 Report
Comments and Suggestions for Authors
The stud by Paskal et al investigated preincisional and multiple intradermal injection of N-acetyl cysteine slightly improves incisional wound healing in an animal model. Although this research study has conducted fairly extensive study on the effect of N-acetyl cysteine on wound healing, the data presented here may not be quite reliable due to design of the experiment. When we look at the fig 8 of this study, this figure clearly shows that wound creation in this study is not consistent across all your groups at least in terms of the length of incisional wounds. Also, created wounds seems to have random distance from each other. In other words, one wound is closer to one wound, but the same wound is a bit far from the other wound. Since this study used different dosage of the NAC, it is essential that wounds are created fairly away from each other to avoid inconsistency in drug treatment and data interpretation and analysis.
Some other comments/suggestion:
· Citation needed for “Our previous studies concern a unique model of preincisional administration of NAC to the future wound bed of a surgical wound, as an additive to local anaesthesia.”
Fig 2:
· It would be good to put n value in the caption
· Having a huge error bar for Control in (EPI) is concerning. This questions the “control” of this study, at least for measuring EPI ( i.e thickness of the epidermis). I suggest doing remeasuring.
· L: doesn’t have Y axis title
Table 1:
What does “n’ in the table stand for? Example: CD31 n of positive cells/mm2. Please mention in the table caption
Fig 4:
· It would be good to put n value in the caption
This study measures extensively skin wound healing indexes. However, in the Materials and Methods section how these measurements are done is lacking. It is highly recommended to provide a separate excel sheet (as a supplement) describing how these measurements were performed.
In assessment of scars we see different time points ( Day 3, day 7, etc) presented by the authors; however, it is not clear when the authors considered the scar as day 0 or day 1? It would be good to have some clarifications on when a wound scar is called scar.
Please put n values in all the figure captions.
Fig 6: Please explain what could be a reason to see a drop in reepithelialisation in NAC 15
2.5 subheading title. First half is italic but the rest is not. P
Fig 7 B: why n value for your control is much higher than your treatment groups? Please explain as this greatly impacts your data interpretation and statistics outcomes.
Comments on the Quality of English LanguageThis manuscript requires professional proofreading to improve its scientific accuracy and ensure clarity and coherence throughout
Round 2
Reviewer 1 Report
Comments and Suggestions for Authors
Accept
Reviewer 2 Report
Comments and Suggestions for Authors
The quality of the manuscript has been greatly improved. All the questions/concerns have been sufficiently addressed.